# Secondary Volatile Metabolite Composition in *Scorzonera pseudolanata* Grossh. Plant Parts

**DOI:** 10.3390/plants14111624

**Published:** 2025-05-26

**Authors:** Aysel Özcan Aykutlu, Serdar Makbul, Kamil Coşkunçelebi, Fatih Seyis

**Affiliations:** 1Field Crops Department, Faculty of Agriculture, Recep Tayyip Erdoğan University, 53300 Pazar-Rize, Türkiye; fatih.seyis@erdogan.edu.tr; 2Biology Department, Faculty of Science, Recep Tayyip Erdoğan University, 53100 Rize, Türkiye; serdar.makbul@erdogan.edu.tr; 3Biology Department, Faculty of Science, Karadeniz Technical University, 61080 Trabzon, Türkiye; kamil@ktu.edu.tr

**Keywords:** *Scorzonera*, chemical content, secondary volatile metabolite, SPME

## Abstract

*Scorzonera* species exhibit various biological activities largely dependent on their chemical composition. While numerous studies have investigated these species’ secondary volatile metabolite content, to our knowledge, this is the first comprehensive study focused on *Scorzonera pseudolanata*. The present study aimed to identify and analyze the secondary volatile metabolites in different parts of *S*. *pseudolanata*. The composition of these metabolites was determined using gas chromatography–mass spectrometry (GC-MS). The resulting data were further analyzed through biplot analysis to differentiate among the plant parts. A total of 46 secondary volatile compounds were identified across all examined tissues. Hexadecane (13.6%) was the dominant compound in the roots, phytone (16.36%) in the leaves, and nonadecane (56.45%) in the seeds. The secondary volatile metabolite profile of *S*. *pseudolanata* differs markedly from that of other *Scorzonera* species, a distinction effectively visualized using a biplot diagram. This study represents the first detailed investigation into the secondary volatile metabolite composition of *S*. *pseudolanata*. It offers foundational data that may inform future in-depth research, thereby contributing to a broader understanding of the phytochemistry of this species.

## 1. Introduction

The genus *Scorzonera* L. is represented by approximately 180–190 species worldwide and constitutes the largest and most widely distributed group within the subtribe Scorzonerinae of the tribe Cichorieae [1,2,3]. Species of the genus *Scorzonera* are found in temperate and tropical regions of Europe and North Africa, as well as arid, alpine, and mountainous environments of the Iranian–Turanian floristic region. Due to long-standing taxonomic challenges, the genus has undergone numerous revisions. The most recent taxonomic revision proposed that *Tourneuxia* Coss.; *Gelasia* Cass.; *Epilasia* (Bunge) Benth.; *Lipschitzia* Zaika, Sukhor & N. Kilian; *Pterachaenia* (Benth.) Stewart; *Koelpinia* Pall.; *Ramaliella* Zaika, Sukhor & N. Kilian; *Pseudopodospermum* (Lipsch. & Krasch.) Kuth.; and *Takhtajaniantha* Nazarova should be recognized as separate genera from *Scorzonera*. Consequently, many taxa formerly classified under *Scorzonera* have been reassigned to these revised genera. Members of the genus *Scorzonera* typically possess a caudex or tuber and are rarely biennial or dwarf subshrubs. Their leaves range from linear to oblong and from entire to pinnatisect. The capitula are characterized by yellow ligules and an involucre of imbricate phyllaries arranged in several series. Achenes may or may not have a tubular carpopodium and are accompanied by a pappus composed of barbellate or plumose bristles [4]. In Türkiye, the genus is represented by 59 taxa, 31 endemic. *Scorzonera* species have a rich history of use in traditional medicine in Anatolia and other regions of the world [5,6,7,8,9,10,11]. In European folk medicine, they treat pulmonary diseases, colds, wounds, and gastrointestinal disorders and are valued for their stomachic, diuretic, galactagogue, antipyretic, and appetite-stimulating properties. Traditional Chinese and Mongolian medicine treats diarrhea, lung edema, parasitic diseases, and fevers caused by bacterial and viral infections [7,11].

*Scorzonera pseudolanata*, a species native to the Iranian–Turanian phytogeographic region, typically grows in the inner, northern, and western regions of Anatolia, Türkiye. The species is characterized by its scapose growth form, with a solitary capitulum, lanceolate-linear leaves, and dense lanate hairs covering all plant parts. It also exhibits yellowish ligules, non-stipitate achenes, and a plumose pappus [5]. These plants prefer alpine steppe meadows and dry calcareous soils. Although *S*. *pseudolanata* is not currently classified in any risk category according to the *Red Data Book of Turkish Plants* [12] and recent assessments [5], it is distributed across a broad geographic area in Anatolia but occurs in limited and localized populations.

Volatile oils exhibit various biological activities, including insecticidal, antiviral, antioxidant, and antibacterial properties [13]. In addition to their role in cancer treatment, they are employed as natural organic compounds and pharmaceuticals. They are widely used in food preservation, aromatherapy, wound healing, and perfumery [14,15,16]. The significance of volatile oils continues to grow due to their diverse applications across various sectors, including the beverage and food industries and the cosmetics and fragrance industries, particularly in producing high-value perfumes with demonstrated bioactivity [17].

Over the years, numerous studies have investigated various Scorzonera species’ chemical composition and biological activities. A comprehensive review of traditional uses, phytochemistry, pharmacology, toxicology, chemotaxonomy, and other applications was published [18]. Table 1 summarizes the published research on the chemical profiles of different *Scorzonera* species and their associated biological activities.

The literature shows that no research on the secondary volatile metabolite content of *Scorzonera pseudolanata* in Türkiye is currently available. Significant differences were observed in the metabolite profiles among the various plant parts, highlighting the chemical diversity within the species.

## 2. Results

Table 2 and Figure 1 show the distribution of secondary volatile metabolite composition in plant parts of *S*. *pseudolanata*.

As shown in Table 2, 46 distinct secondary volatile metabolite compounds were identified across all plant parts of Scorzonera pseudolanata. In the root, 22 compounds accounted for 100.00% of the volatile content; in the leaf, 35 compounds represented 99.9%; and in the seed, 30 compounds comprised 99.66% of the volatile oil profile.

Six major chemical groups were identified: (1) oxygenated monoterpenes; (2) oxygenated sesquiterpenes; (3) sesquiterpene hydrocarbons; (4) alcohols, ketones, aldehydes, and furans; (5) alkanes, alkenes, alkynes, and arenes; and (6) ethers, carboxylic acids, and esters (Table 2 and Figure 1). Among these, the group of alkanes, alkenes, alkynes, and arenes was dominantly present in all plant parts analyzed.

The dominant chemical compound in the root was hexadecane (13.6%), in the leaf phytone (16.36%), and in the seed nonadecane (56.45%) (Table 2). Seven compounds were exclusively detected in the root: isoborneol (1.94%), capraldehyde (1.90%), cyclosativene (5.81%), linalyl benzoate (7.26%), civetone (4.06%), geranyl butyrate (1.82%), and dodecalactone (2.76%). In contrast, 11 components were uniquely identified in the leaf, including eucalyptol (2.00%), α-copaene (2.40%), β-bourbonene (1.95%), α-humulene (1.90%), β-ionone (1.88%), δ-cadinene (1.57%), theaspirane (1.42%), tridecylaldehyde (0.95%), phytone (16.36%), and hexadecanoic acid (1.02%). Only three compounds were detected exclusively in the seed: caprylaldehyde (0.17%), undecalactone (0.16%), and myristate (0.16%).

Figure 2 presents a biplot analysis based on 10 selected secondary volatile metabolites with concentrations exceeding 6%, including β-caryophyllene, pentadecanol, linalyl benzoate, hexadecane, octadecane, Nonadecane, heneicosane, phytone, and methyl palmitate.

The total variation observed was fully explained (100%) by the first two principal components derived from the analysis. The leaf samples were clearly distinguished from the root and seed samples, primarily due to the influence of β-caryophyllene and phytone. Conversely, nonadecane and heneicosane were the key compounds that differentiated the seed samples from the root and leaf samples.

## 3. Discussion

A total of 47 distinct secondary volatile metabolite components were identified across the different plant parts of *Scorzonera pseudolanata*. However, the proportion and distribution of these compounds varied significantly among the root, leaf, and seed tissues. In the present study, the secondary volatile metabolite profiles of the examined plant parts were distinguishable, as demonstrated in Figure 1 and Figure 2.

Endogenous factors (such as plant physiology and function) and environmental conditions (including light, precipitation, soil characteristics, and overall growth environment) can contribute to the variation in volatile oil composition. These variables collectively contribute to differentiating chemical profiles among plant parts [53].

Only a few studies have examined other Scorzonera species’ secondary volatile metabolite composition. For instance, in *S*. *hispanica*, collected from Germany, hexadecanoic acid (20.3%) was identified as the most abundant compound, followed by octane (7.5%), hexane (4.8%), and octadecanoic acid (3%) [7]. In the roots of *S*. *undulata* ssp. *deliciosa* (Guiss.) Maire, hexadecanoic acid (42.2%), *n*-tetradecanoic acid (16.1%), octadecanoic acid (7.7%), and hexadecenoic acid (4.5%) were identified as major constituents. Additionally, methyl hexadecanoate (30.4%), methyl linoleate (23.9%), heneicosane (12.2%), and octadecane (4.4%) were reported as dominant compounds in the same subspecies [21].

Trimethyl pentadecanone (27.73%), caryophyllene oxide (16.84%), neophyte diene (7.68%), and (E)-ionone (6.77%) were identified in the oil extracted from the leaves and flowers of *Scorzonera calyculata* [54]. Oxygenated sesquiterpenes constituted the largest proportion of the volatile oil (20.68%), followed by diterpenes (8.34%), monoterpene hydrocarbons (4.75%), sesquiterpene hydrocarbons (1.88%), and oxygenated monoterpenes (1.04%).

In *S*. *sandrasica*, the most abundant compounds were caryophyllene oxide (19.7%), manoyl oxide (16.05%), manool (11.3%), 2-oxo-manoyloxide (8.9%), sclareol (7.7%), and β-caryophyllene (7.6%). Carvacrol accounted for 2.7% of the total volatile oil content [55].

In *S*. *acuminata*, collected from Ankara/Türkiye, α-copaene, β-caryophyllene, β-ionone, capronaldehyde, pelargonaldehyde, pentadecanol, myristic alcohol, tetradecane, pentadecane, hexadecane, heptadecane, octadecane, heneicosane, and phytone were detected in all plant parts (root, stem, leaf, and seed). The highest concentrations were recorded as 27.16% for β-caryophyllene in the root, 12.97% in the stem, 25.96% in the leaf, and 22.02% for lauryl alcohol in the seed [48].

GC-MS analysis of *S*. *papposa*, collected from Erzincan/Türkiye, revealed 56 different volatile oil components in various plant parts. The volatile oil composition showed substantial variation among plant tissues, indicating potential for pharmaceutical applications. The highest concentrations were 16.78% for methyl palmitate in the root, 30.44% for phytone in the stem, 40.17% for phytone in the leaf, and 31.84% for phytol in the seed [49].

Another study investigated the volatile oil compositions of the root, stem, and leaf parts of *S*. *mollis* ssp. *mollis* and *S*. *mollis* ssp. *szowitzii* collected from Tekirdağ/Türkiye [50]. A total of 70 volatile compounds were identified across all plant parts. In *S*. *mollis* ssp. *mollis*, the major constituents included civetone (42.62%) in the root, β-caryophyllene (11.82%) in the stem, and phytol (12.08%) in the leaf. In *S*. *mollis* ssp. *szowitzii*, the highest values were observed for hexadecane (16.42%) in the root, cyclosativene (20.68%) in the stem, and both β-patchoulene and cyclosativene (12.31%) in the leaf. The volatile oil profiles of both species were dominated by sesquiterpene hydrocarbons, alcohols/ketones/aldehydes/furans, and alkanes/alkenes/alkynes/arenes.

These findings collectively suggest that the secondary volatile metabolite composition of *S*. *pseudolanata* is distinct from that of other *Scorzonera species*. Furthermore, the secondary metabolites identified in *S*. *pseudolanata* have notable biological activities.

Eucalyptol, an oxygenated monoterpene detected in the leaf, is widely used in traditional medicine and occurs naturally in *Eucalyptus*, *Rosmarinus*, and *Cinnamomum camphora*. It has demonstrated anti-inflammatory, antioxidant, antimicrobial, bronchodilatory, analgesic, and pro-apoptotic properties [56]. Isoborneol exhibits potent antiviral activity, particularly against the HSV-1 virus [57], and decanal is known for its antioxidant effects [58].

Caryophyllene oxide, one of the oxygenated sesquiterpenes identified, functions as a broad-spectrum antifungal agent in plant defense and has insecticidal and antifeedant properties [59,60]. Alpha-santalol has been shown to possess antitumor and cancer-preventive effects [61]. Cyclosativene is found only in the root of *S. pseudolanata* and *displays antioxidant* and anticarcinogenic properties [62]. Alpha-copaene (α-COP), present in many medicinal and aromatic plant volatile oils, exhibits antioxidant and antigenotoxic properties [63]. Beta-bourbonene is known for its anti-inflammatory and antioxidant effects and is used as a preventive agent [64]. β-Caryophyllene, present in all parts of *S*. *pseudolanata*, has been reported to possess antibacterial, antioxidant, gastroprotective, anxiolytic, and anti-inflammatory activities [65]. Additionally, α-humulene displays antitumor, anti-inflammatory, and antimicrobial properties [66], and germacrene D has notable antibacterial activity [67,68]. β-Ionone, detected exclusively in the leaf, is a multifunctional compound widely distributed in flowers, fruits, and vegetables. It contributes to flavors, aromas, pigments, growth regulation, and ecological interactions, including insect attraction or repellence, and has antibacterial and fungicidal properties [69]. Furthermore, δ-cadinene has demonstrated antimicrobial activity [70].

Further, several compounds with important biological functions were detected: capronaldehyde (antibacterial) [71], caprylaldehyde (antiviral) [72], phenylacetaldehyde (antimicrobial and DMPD^+^-scavenging) [73], pelargonaldehyde (antidiarrheal and antimicrobial against Gram-positive and Gram-negative bacteria, and antifungal) [74,75], theaspirane (antioxidant) [76], lauryl alcohol (bactericidal and fungicidal) [77], tridecylaldehyde (bactericidal and antibacterial) [71,78], tetradecanal (antibacterial) [79], myristic alcohol (antibacterial and anti-inflammatory) [80], pentadecanol (antibacterial) [81], linalyl benzoate (antimicrobial) [82], civetone (noted for its medicinal potential) [83], and phytol (antioxidant, anti-inflammatory, and antimicrobial) [84].

Alkanes, alkenes, alkyne, and arenes formed the biggest chemical group in *S. pseudolanata* plant parts (Table 2). Tetradecane is known for its antibacterial and antifungal activity [85]; pentadecane for antimicrobial activity [86,87]; hexadecane for antifungal, antibacterial, and antioxidant activity [88,89]; heptadecane for antioxidant activity [90]; octadecane for antifungal, anti-influenza, antimicrobial, anti-inflammatory, and antioxidant properties [91]; nonadecane for antimicrobial activity, anti-HIV, and antioxidants [92]; eicosane for antifungal activity [93]; and heneicosane for antibacterial and antifungal activities [94].

Detected components belonging to the ethers, such as carboxylic acids and esters, also have important biological activities. For example, geranyl acetate shows antibacterial activity [95]; geranyl butyrate shows anticancer activity [96]; citronellyl butyrate shows antibacterial activity [97]; undecalactone shows antimicrobial activity [98]; dihydrojasmonate shows anticancer activity [99]; myristate shows repellent activity [100]; dodecalactone shows antifungal activity [101]; apiole shows antitumor activity [102]; phytone shows anti-inflammatory properties [103,104,105,106]; hexadecenoic acid shows antioxidant, hypocholosterolomic, nematicidal, and pesticidal activity [107]; methyl palmitate shows antifungal and antioxidant activity [108]; and geranyl benzoate antifungal and antioxidant activity [109].

As demonstrated, the different secondary volatile oil components identified in various parts of *Scorzonera pseudolanata* exhibit numerous biologically active properties of medicinal relevance. It is important to note that the present findings are based on samples collected from natural habitats. To better utilize these bioactive compounds, further studies are required to focus on the propagation, agronomic practices, harvesting, and isolation of specific components in *S*. *pseudolanata*.

Analytical tools such as biplot analysis are valuable for identifying genotypes and grouping based on chemical similarity [110,111]. Biplot analysis facilitates the differentiation of plant materials and can effectively distinguish species according to their chemical profiles [112,113]. Combined with other analytical approaches, it can also aid in identifying traits critical to genetic variability in crop species [114]. In a biplot, variables contributing to the distinction between different variants can be visualized and categorized [115]. In this study, root, leaf, and seed parts of *S*. *pseudolanata* were differentiated based on their secondary volatile metabolite composition using such chemical data.

The Asteraceae family is one of the largest flowering plants worldwide and is known for containing a wide array of biologically active chemical constituents [116,117]. The *Scorzonera* genus, a member of this family, includes species widely used in food and traditional medicine in Türkiye. There remains a strong need for further investigation into the biological activities of the genus’s chemical constituents to enrich the scientific literature on plant-derived pharmaceuticals.

## 4. Materials and Methods

*S*. *pseudolanata* specimens were collected from the open steppe areas surrounding the Köse district of Gümüşhane, located in the northern region of Türkiye, at elevations ranging from approximately 1600 to 1700 m (Figure 3). Photographs of the plant’s natural habitat and overall morphology are provided in Figure 1. Voucher specimens (Makbul 526 & Coşkunçelebi) have been deposited in the Herbarium of the Department of Biology at Recep Tayyip Erdoğan University (RUB), Rize, Türkiye. The additional plant materials used for chemical analysis were air-dried under ambient conditions at room temperature

### 4.1. Sample Preparation

The instrumental system used for the analysis comprised a Shimadzu GC-2010 Plus gas chromatograph, a QP2020 mass spectrometer, and a multifunctional autosampler (AOC-5000 Plus/SHIMADZU) equipped with a solid-phase microextraction (SPME) module and a split/splitless injection inlet.

The SPME extraction was performed using a polydimethylsiloxane (PDMS) fiber (1 cm × 100 μm thickness) purchased from Sigma-Aldrich (Supelco, Bellefonte, PA, USA). The fiber was preconditioned for 5 min at 250 °C before analysis and reconditioned for 10 min at 250 °C after each run.

For each sample, 1.0 g of plant material was placed in a 20 mL SPME glass vial, followed by the addition of 100 μL of hexane. The same procedure was applied to all samples, which were sealed with silicone/PTFE septa. Volatile compounds were extracted under optimized headspace SPME (HS-SPME) conditions, consisting of a 5 min equilibration period and a 15 min extraction at 100 °C with an agitation speed of 500 rpm. Analytes were desorbed for 1 min in the GC injection port at 250 °C using a straight Ultra Inert SPME liner operating in split mode. Each sample was analyzed in duplicate.

### 4.2. Secondary Metabolite Composition Analysis

All analyses employed a Shimadzu GC-2010 Plus gas chromatograph and a QP2020 mass selective detector. The separation procedure used an Rtx-5MS low-bleed capillary column (30 m × 0.25 mm × 0.25 μm; Restek, Bellefonte, PA, USA). Helium served as the carrier gas at a constant pressure of 80 kPa. The oven temperature program was optimized as follows: initial temperature of 40 °C (held for 2 min), increased to 250 °C at a rate of 4 °C/min, and held at 250 °C for 3 min, resulting in a total run time of 55 min. The interface and ion source temperatures were set at 250 °C and 200 °C, respectively. Electron ionization (EI) was conducted at 70 eV with an *m*/*z* scan range of 40–500 and a scan speed of 1666 u/s.

Data acquisition and processing were carried out using GCMS solution software Version 4.53 (Shimadzu, Japan). Compound identification was performed by comparing mass spectra to the Wiley FFNSC 3rd Edition Library (Mass Spectra of Flavours and Fragrances of Natural and Synthetic Compounds) and by matching calculated retention indices (RIs) with those listed in the FFNSC 3rd Edition Library, the NIST Chemistry WebBook (SRD 69), and the PubChem database.

The Alkan standard was analyzed using the same method. The index values of substances were calculated by entering the RI values of alkanes into the software. C7–C30 saturated alkanes were used as certified reference material; 1000 μg/mL each component was solved in hexane. Using the present method, RI was calculated using the Sigma-Aldrich standard. Afterwards, the analysis of C7–C30 was conducted.

To calculate the compound percentages, the percentage peak area method was used. This method uses the area of the target component (component A) peak as a proportion of the total area of all detected peaks to analyze quantity.

### 4.3. Data Analysis

XLSTAT 2024 analysis software (Lumivero) was used to perform Hierarchical Cluster Analysis (HCA) to visualize the chemical variability among the different plant parts of *Scorzonera pseudolanata*. The obtained data were also used to construct a biplot diagram to illustrate further the distribution and differentiation based on secondary volatile metabolite composition [118].

## 5. Conclusions

This study represents the first investigation into the secondary volatile metabolite composition of *Scorzonera pseudolanata*. The present results revealed notable differences in the secondary metabolite profiles among the various plant parts of this species. However, further research should validate these findings using additional *S*. *pseudolanata* samples.

The data presented here provide a valuable foundation for more comprehensive future studies and contribute to a deeper understanding of the chemical composition of *S*. *pseudolanata*.

## Figures and Tables

**Figure 1 plants-14-01624-f001:**
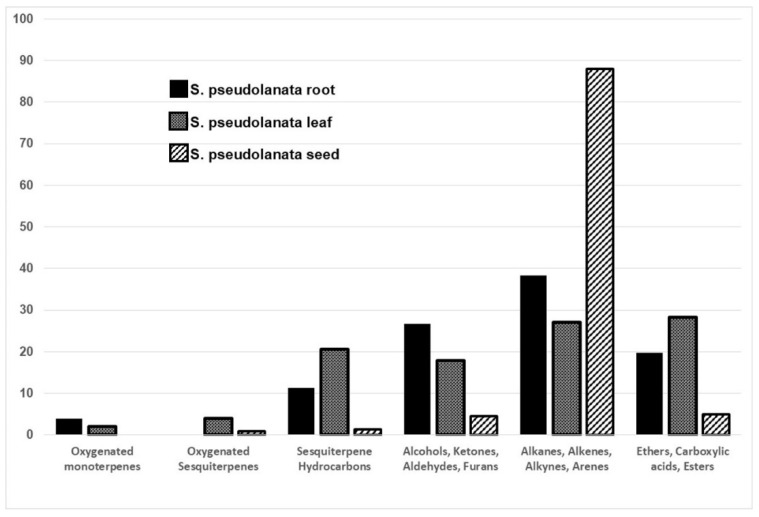
Distribution of chemical groups in different plant parts of *Scorzonera pseudolanata*.

**Figure 2 plants-14-01624-f002:**
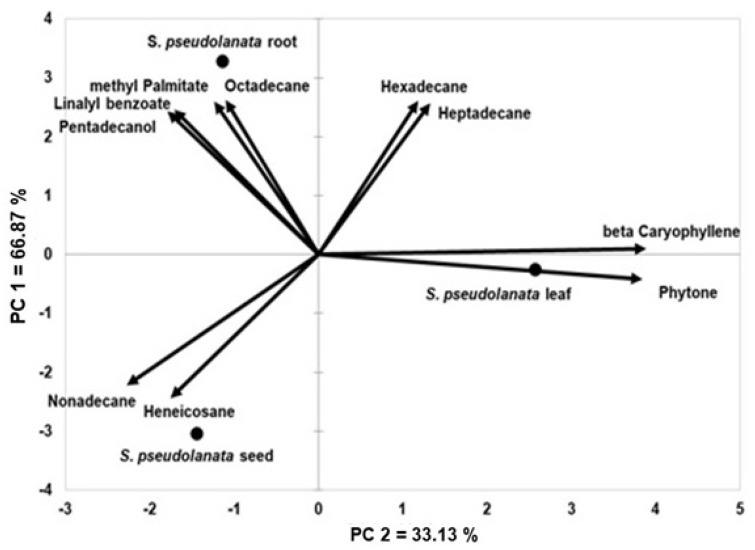
Distribution of composition of volatile secondary metabolites in the roots, leaves, and seeds of *S*. *pseudolanata*.

**Figure 3 plants-14-01624-f003:**
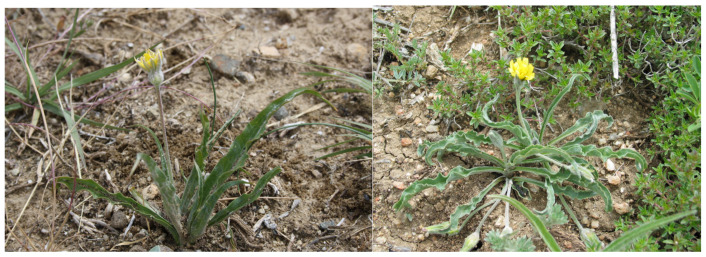
*Scorzonera pseudolanata* (Makbul 526 & Coşkunçelebi).

**Table 1 plants-14-01624-t001:** Research on *Scorzonera* species based on chemical content.

Investigated Components	Investigated Part	Species	Reference
Phenolic compounds	Subaerial parts	*S*. *tomentosa* Siev. ex Ledeb.	[19]
Volatile secondary metabolite composition and phenolic content	Roots [20], capitula and aerial parts [21], aerial parts [22,23],	*S*. *Scorzonera undulata* ssp. *deliciosa* (Guiss) Maire., *Scorzonera undulata* Vahl, *Scorzonera sandrasica* Hartvig & Strid, *Scorzonera calyculata* Boiss.	[20,21,22,23]
Phenolic content	Flowering parts	*S*. *undulata* ssp. *deliciosa*, *S*. *undulata*, *S*. *sandrasica*, *S*. *calyculata* Boiss.	[24]
Chemosystematic studies	Aerial parts [25], subaerial parts [26]	*S*. *cinerea* Boiss., *S*. *incisa* DC., *S*. *eriophora* DC., *S*. *laciniata* Jacq., *S*. *parviflora* Jacq., *S*. *cana* (C.A. Meyer) Hoffm. *var*. *alpina* (Boiss.) D.F.Chamb., *S*. *cana* (C.A. Meyer) Hoffm. var. *jacquiniana* (*W*.*Koch*) D.F.Chamb.	[25,26]
Phenolic components and in vitro antioxidant, antibacterial, anti-inflammatory	Roots and leaves	*S*. *austriaca* Balb., *S*. *aristata* Ramond ex DC., *S*. *montana* var. *boetica* Boiss. ex DC., *S*. *hispanica* L., *S*. *crispatula* Boiss., *S*. *trachysperma* Günther ex Spreng., *S*. *villosa* Scop.	[18]
Phenolic compounds	Subaerial parts [19], roots [27], aerial parts and roots [28]	*S*. *hieraciifolia* Hayek	[19,27,28]
Phytochemical components and antioxidant activity	Aerial parts	*S*. *judaica* Eig and *S*. *tomentosa*	[29]
Chemical composition	Dried roots [20], aerial parts [21,22], whole plant [30]	*S*. *sandrasica*, *S*. *undulata*, *S*. *undulata* ssp. *deliciosa*, *S*. *hispanica*	[20,21,22,30]
Antioxidant and antihyperglycemic activity	Leaves	*S*. *cinerea*	[31]
Chemical components (GC-MS), antioxidant, anticancer and antibacterial activity	Aerial parts	*S*. *calyculata*	[23]
Anti-antinociceptive action and natural compounds	Leaves and rootstocks	*S*. *latifolia* DC., *S*. *mollis* ssp. *szowitzii* Chamb., *S*. *suberosa* K.Koch, *S*. *tomentosa*, *S*. *aristata*	[32]
Biologically active natural compounds	Aerial parts	*S*. *divaricata* Aucher ex DC., *S*. *pseudodivaricata* Lipsch.	[10]
Prospective neurobiological effect	Aerial parts and roots	27 different *Scorzonera* species including *S*. *pseudolanata* Grossh.	[33]
Phenolic compounds and certain biological activities	Aerial parts and roots	*S*. *pygmaea* Sm.	[34]
Antidiabetic effects	Aerial parts	*S*. *tomentosa*, *S*. *mollis* ssp. *szowitzii*, *S*. *suberosa*, *S*. *eriophora*, *S*. *acuminata* Boiss., *S*. *sublanata* Lipsch., *S*. *cana* var. *jacquiniana*	[35]
Pharmacognostic, antibacterial, and laxative investigation	Aerial parts and roots	*S*. *undulata*	[36]
Phenolic compounds	Aerial parts	*S*. *aristata*, *S*. *austriaca*, *S*. *montana* var. *boetica*, *S*. *crispatula*, *S*. *hispanica*, *S*. *trachysperma*, and *S*. *villosa*	[25]
Inulin	Roots and leaves	*S*. *hispanica*	[37]
Antibacterial and antibiofilm activity	Whole plant, flowers, stems, leaves and roots	*S*. *mackmeliana* Boiss.	[38]
Antibacterial potential	Whole plant	*S*. *undulata*	[39]
Wound healing	Aerial parts and roots Aerial parts and roots	*S*. *cinerea*, S. *latifolia*, *S*. *incisa*, *S*. *mollis* ssp. *szowitzii*, *S*. *parviflora*, S. *tomentosa* *S*. *acuminata*, *S*. *cana* var. *alpina*, *S*. *cana* var. *jacquiniana*, *S*. *cana* (C.A Meyer) Hoffm. var. *radicosa* (Boiss.) Chamberlain, *S*. *eriophora*, *S*. *suberosa* and *S*. *sublanata*	[40,41]
Fatty acid compositions, chemical content, and antioxidant activity	Leaves and roots	*S*. *paradoxa* Fisch and C.A. Mey	[42]
Chemical constituents	Roots	*Scorzonera divaricata*	[43]
Phytochemical profile and biological Activities	Aerial parts and roots	*S*. *sandrasica*, *S*. *coriacea* A. Duran and Aksoy, and *S*. *ahmet-duranii* Makbul and Coskuncelebi	[44]
Chemical profiles and pharmacological effects	Aerial parts and roots	*S*. *hieraciifolia*, *S*. *hispanica*, *S*. *tomentosa*	[45]
Biological activity	Leaves	*S*. *tomentosa*	[46]
Phenolics, terpenoids, and potential bioactivities	Aerial parts	*S*. *incisa*	[47]
Secondary volatile metabolite composition and phenolic content	Root, stem, leaf, and seed	*S*. *acuminata*	[48]
	Root, stem, leaf, and seed	*S*. *papposa* DC.	[49]
	Root, stem, and leaves	*S*. *mollis* M.Bieb. ssp. *mollis and S*. *mollis* ssp. *szowitzii*	[50]
Phenolic content	Root, stem, leaf, and seed	25 *Scorzonera* species, including *S*. *pseudolanata*	[51]

**Table 2 plants-14-01624-t002:** The secondary volatile metabolite components detected in *S*. *pseudolanata* plant parts (%).

No	RI *	RI in Library **	Component	Root			Leaf			Seed		
				Area %	Measured RI Value	Similarity %	Area %	Measured RI Value	Similarity %	Area %	Measured RI Value	Similarity %
1	801	801	Capronaldehyde	2.38	801	96	1.73	801	96	0.39	801	97
2	1003	1006	Caprylaldehyde	-	-	-	-	-	-	0.17	1008	94
3	1032	1032	Eucalyptol	-	-	-	2.00	1032	95	-	-	-
4	1042	1045	Phenylacetaldehyde	3.51	1043	95	1.08	1043	95	0.3	1043	97
5	1107	1107	Pelargonaldehyde	-	-	-	2.33	1105	96	0.52	1105	97
6	1167	1165	Isoborneol	1.94	1168	94	-	-	-	-	-	-
7	1206	1208	Decanal	1.90	1211	89	-	-	-	-	-	-
8	1367	1367	Cyclosativene	5.81	1380	89	-	-	-	-	-	-
9	1375	1375	α Copaene	-	-	-	2,40	1376	95	-	-	-
10	1389	1382	β Bourbonene	-	-	-	1.95	1385	95	-	-	-
11	1400	1400	Tetradecane	3.05	1402	89	1.29	1402	97	0.32	1402	98
12	1418	1424	β Caryophyllene	2.18	1456	91	8.28	1425	98	1.39	1425	97
13	1454	1450	Geranyl acetone	-	-	-	2.30	1452	96	0.17	1452	98
14	1456	1459	Geranyl butyrate	1.82	1462	92	-	-	-	-	-	-
15	1458	1454	α Humulene	-	-	-	1.90	1456	98	-	-	-
16	1465	1447	Theaspirane	-	-	-	1.42	1449	93	-	-	-
17	1485	1480	Germacrene D	3.34	1484	92	2.63	1484	92	-	-	-
18	1490	1490	β Ionone	-	-	-	1.88	1490	92	-	-	-
19	1493	1476	Lauryl alcohol	-	-	-	1.87	1477	96	1.88	1477	96
20	1500	1500	Pentadecane	3.94	1502	90	1.69	1502	96	0.39	1502	98
21	1510	1516	Tridecylaldehyde	-	-	-	2.23	1519	94	0.18	1518	95
22	1529	1518	δ Cadinene	-	-	-	1.57	1520	90	-	-	-
23	1532	1529	Citronellyl butyrate	-	-	-	2.44	1530	99	0.27	1530	99
24	1577	1602	Undecalactone	-	-	-	-	-	-	0.16	1603	97
25	1589	1587	Caryophyllene oxide	-	-	-	1.60	1588	92	0.73	1588	91
26	1600	1600	Hexadecane	13.06	1602	95	9.76	1602	98	152	1602	95
27	1615	1614	Tetradecanal	-	-	-	1.82	1615	97	0.5	1615	93
28	1620	1708	Dodecalactone	2.76	1709	9	-	-	-	0.3	1709	98
29	1657	1653	Dihydrojasmonate	-	-	-	0.97	1658	95	0.27	1658	91
30	1666	1727	Myristate	-	-	-	-	-	-	0.16	1729	97
31	1671	1676	α Santalol	-	-	-	2.44	1678	95	0.16	1678	92
32	1687	1683	Apiole	3.78	1687	87	2.77	1687	94	0.25	1687	
33	1695	1680	Myristic alcohol	2.79	1682	96	1.48	1682	95	0.18	1682	95
34	1700	1700	Heptadecane	6.55	1702	97	5.37	1702	96	1.83	1702	98
35	1784	1784	Pentadecanol	6.73	1786	91	-	-	-	-	-	-
36	1792	1796	Linalyl benzoate	7.26	1798	97	-	-	-	-	-	-
37	1800	1800	Octadecane	8.11	1802	97	2.66	1802	98	1.46	1802	99
38	1841	1841	Phytone	-	-	-	16.36	1848	99	1.57	1848	97
39	1901	1900	Nonadecane	3.65	1903	92	1.69	1903	90	56.45	1903	98
40	1922	1977	Hexadecenoic acid	-	-	-	1.02	1978	94	-	-	-
41	1925	1925	methyl Palmitate	6.75	1929	93	2.47	1929	97	1.75	1929	96
42	1972	1968	Geranyl benzoate	4.63	1968	94	-	-	-	-	-	-
43	2001	2000	Eicosane	-	-	-	1.49	2002	95	2.19	2002	98
44	2020	2016	Civetone	4.06	2020	95	-	-	-	-	-	-
45	2100	2100	Heneicosane	-	-	-	3.1	2103	95	23.82	2103	98
46	2115	2106	Phytol	-	-	-	1.29	1402	97	0.32	1402	98

* Kovats Retention Index (RI); ** [52].

## Data Availability

The original contributions presented in this study are included in the article. Further inquiries can be directed to the corresponding author.

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
