# Peer review of "Secondary Volatile Metabolite Composition in Scorzonera pseudolanata Grossh. Plant Parts"

_plants, 2025, doi:10.3390/plants14111624_

Round 1
Reviewer 1 Report
Comments and Suggestions for Authors📝 Reviewer Comments – Plants Manuscript ID: plants-3579463-peer-review-v1
Title: Secondary volatile metabolite composition in Scorzonera pseudolanata Grossh. plant parts
General Evaluation
This manuscript presents the first report on the secondary volatile metabolite profile of Scorzonera pseudolanata, using HS-SPME/GC-MS to analyze root, leaf, and seed parts. The work is of potential value in phytochemical and taxonomic research; however, the section on compound identification requires substantial improvement to meet the standards of scientific rigor.
Major Concerns
- Compound Identification Accuracy
The study relies solely on identification via the Shimadzu FFNSC library, without providing:
- Match scores or confidence intervals for spectral matches.
- Confirmation with authentic standards.
- Retention index (RI) matching with literature values, although RI values were measured.
Recommendation: Please include mass spectral similarity indices and, where possible, compare your Kovats RI data to published standards. Confirmation with authentic reference standards for major compounds (e.g., phytone, nonadecane, β-caryophyllene) would greatly enhance the validity of your results.
- Co-elution and Peak Resolution
The manuscript does not describe how co-elution or overlapping peaks were addressed. HS-SPME analysis of complex plant matrices often suffers from this issue, which can significantly compromise compound identity.
Recommendation: Include chromatographic profiles (TIC/EICs) in supplementary material and describe any use of deconvolution software (e.g., AMDIS) to validate peak purity and assign identities confidently.
- Quantification Approach
The reported percentages are derived from relative peak area normalization, yet there is no mention of:
- Use of internal standards or
- Calibration procedures.
While this is acceptable for qualitative screening, please clarify that the data are semi-quantitative and interpret results accordingly. If possible, consider adding an internal standard in future work to improve reproducibility.
- Chemical Group Classification
The classification of compounds into broad categories (e.g., “Alcohols, Ketones, Aldehydes, Furans”) is too general and at times chemically inconsistent. For instance, phytol (a diterpenoid alcohol) is grouped with phenylacetaldehyde, an aromatic aldehyde.
Recommendation: Consider revising the classification to reflect biosynthetic origin or compound class more precisely (e.g., terpenoids, aliphatic hydrocarbons, benzenoids), which would help readers better understand the chemotaxonomic or pharmacological implications.
- Biological Activity Claims
The manuscript lists many biological activities based on literature for each identified compound. However, several of these claims are based on unconfirmed identities, low concentrations, or distantly related biological models.
Recommendation: Please moderate these discussions, focusing only on compounds identified with high confidence and present in meaningful concentrations. Emphasize that the biological relevance remains hypothetical unless validated by bioassays or targeted isolation.
Minor Comments
- Please clarify whether identification relied on both mass spectra and RI values, or only the spectral library match.
- Figures and tables would benefit from standardization of units and precision (e.g., consistent decimal places).
- Consider rewording the conclusion to reflect that compound identification was preliminary, pending further validation.
Conclusion
This manuscript has potential to contribute to the chemotaxonomic understanding of Scorzonera species. However, I recommend major revision, with a focus on strengthening the compound identification methodology, clarifying the quantitative interpretation, and tempering conclusions regarding bioactivity.
I look forward to reviewing a revised version that addresses these concerns.

Author Response
Reviewer Comment
General Evaluation
This manuscript presents the first report on the secondary volatile metabolite profile of Scorzonera pseudolanata, using HS-SPME/GC-MS to analyze root, leaf, and seed parts. The work is of potential value in phytochemical and taxonomic research; however, the section on compound identification requires substantial improvement to meet the standards of scientific rigor.
Major Concerns
- Compound Identification Accuracy
Answer:
The study relies solely on identification via the Shimadzu FFNSC library, without providing:
We are using the following library:
Mass Spectra of Flavors and Fragrances of Natural and Synthetic Compounds, 3rd Edition (FFNSC 3)
As konwn, there are records for “flavor and fragrance” molecules inthis library and these areÇ
- Spectral Records: 3,462
- Chemical Structures: 3,462
- RI1 = measured on SLB-5MS (Hydro):3,462
- RI2 = measured on SLB-5MS (FAMEs): 2,516
- RI3 = measured on Supelcowax-10 (FAMEs): 1,466 (same records as RI4)
- RI4 = measured on Supelcowax-10 (FAEEs): 1,466 (same records as RI3)
- RI5 = measured on Equity-1 (Hydro): 646
The relevant web adress is given below.
https://www.wiley.com/en-us/Mass+Spectra+of+Flavors+and+Fragrances+of+Natural+and+Synthetic+Compounds%2C+3rd+Edition-p-9781119069843
The used match scores are given in the Table below
Match scores or confidence intervals for spectral matches.
- Confirmation with authentic standards.
We compared similartiy indexes between Dodecane, Tetradecane, Hexadecane standarts and Retention İndex and corresponding library values. (See Table above)
- Retention index (RI) matching with literature values, although RI values were measured.
Recommendation: Please include mass spectral similarity indices and, where possible, compare your Kovats RI data to published standards. Confirmation with authentic reference standards for major compounds (e.g., phytone, nonadecane, β-caryophyllene) would greatly enhance the validity of your results.
C7 - C30 Saturated Alkanes certified reference material was used, 1000 μg/mL of each component was solved in hexane referring Sigma-Aldrich Standard and RI was calculted in present method. Then the detection of C7-C30 was performed.
- Co-elution and Peak Resolution
The manuscript does not describe how co-elution or overlapping peaks were addressed. HS-SPME analysis of complex plant matrices often suffers from this issue, which can significantly compromise compound identity.
Because, the AOC-5000 plus SHIMADZU device was used, and the groups we were specifically interested were extracted from the complex matrix using the appropriate fiber for our analysis They were simplified from the matrix in this way. The column was heated very slowly using the appropriate the GC column temperature program and thereby a clear separation of the analytes was achieved.
Recommendation: Include chromatographic profiles (TIC/EICs) in supplementary material and describe any use of deconvolution software (e.g., AMDIS) to validate peak purity and assign identities confidently.
The related chromatogramme is given in the supplementary matarial part.
- Quantification Approach
The reported percentages are derived from relative peak area normalization, yet there is no mention of:
- Use of internal standardsor
- Calibration procedures.
While this is acceptable for qualitative screening, please clarify that the data are semi-quantitative and interpret results accordingly. If possible, consider adding an internal standard in future work to improve reproducibility.
Related information is given below.
- Chemical Group Classification
The classification of compounds into broad categories (e.g., “Alcohols, Ketones, Aldehydes, Furans”) is too general and at times chemically inconsistent. For instance, phytol (a diterpenoid alcohol) is grouped with phenylacetaldehyde, an aromatic aldehyde.
Recommendation: Consider revising the classification to reflect biosynthetic origin or compound class more precisely (e.g., terpenoids, aliphatic hydrocarbons, benzenoids), which would help readers better understand the chemotaxonomic or pharmacological implications.
The group of Aldehydes, Ketones, Alcohols and Furans were seperated in different groups to giev an better understanding. But i have to mention that we have used the same classification inthe following publications.
- Yurteri E., Makbul S., Gültepe M., Seyis F. (2023) Determination of the chemical composition in different plant parts of S. mollis taxa. Studia Universitatis Babes-Bolyai Chemia, cilt.68, sa.3, ss.161-178, 2023 (SCI-Expanded)
- Yurteri E., Makbul S., Çoşkunçelebi K., Gultepe M., Seyis F. (2022) Evaluation of the Chemical Composition in Different Plant Parts of Scorzonera papposa. Fresenıus Envıronmental Bulletın, cilt.31, sa.3A, ss.3460-3468, 2022 (SCI-Expanded)
- Yurteri E., Makbul S., Çoşkunçelebi K., Seyis F. (2021). Essential Oil Composition in Different Plant Parts of Scorzonera acuminata. NEW DEVELOPMENT ON MEDICINAL AND AROMATIC PLANTS, Özyazıcı Gülen, Editör, Nobel Yayınevi, Ankara, ss.243-264, 2021
Also, Tridecylaldehyde and Tridecanal are heame component. This was also corrected in the Table 1. Therefore the number of detected compunds was corrected as 46 in the manscript.
- Biological Activity Claims
The manuscript lists many biological activities based on literature for each identified compound. However, several of these claims are based on unconfirmed identities, low concentrations, or distantly related biological models.
Recommendation: Please moderate these discussions, focusing only on compounds identified with high confidence and present in meaningful concentrations. Emphasize that the biological relevance remains hypothetical unless validated by bioassays or targeted isolation.
The biological activity of every deteced compound is given to underline teh importance of detected compounds. To underline this Conclusion was rewritten as “This study represents the first investigation into the secondary volatile metabolite composition of Scorzonera pseudolanata. The preliminary results revealed notable differences in the secondary metabolite profiles among the various plant parts of this species. However, further research should validate these findings using additional S. pseudolanata samples. The data presented here provide a valuable foundation for future, more comprehensive studies and contribute to a deeper understanding of the chemical composition of S. pseudolanata."
Minor Comments
- Please clarify whether identification relied on both mass spectra and RI values, or only the spectral library match.
- Are given in supplemenraty material.
- Figures and tables would benefit from standardization of units and precision(e.g., consistent decimal places).
- İn Table and Figures two deciaml places are used.
- Consider rewording the conclusion to reflect that compound identification was preliminary, pending further validation.
- Was changed as “This study represents the first investigation into the secondary volatile metabolite composition of Scorzonera pseudolanata. The preliminary results revealed notable differences in the secondary metabolite profiles among the various plant parts of this species. However, further research should validate these findings using additional pseudolanata samples. The data presented here provide a valuable foundation for future, more comprehensive studies and contribute to a deeper understanding of the chemical composition of S. pseudolanata.

Reviewer 2 Report
Comments and Suggestions for Authors
The manuscript reports the volatile secondary metabolites occurring in different parts of the plant S. pseudolanata harvested in Turkey. Extraction was performed by SPME at room temperature followed by GC-MS analysis. The novelty and originality of this manuscript is fair, but there are interesting data to be a contribution to the chemotaxonomy of S. pseudolanata.
The manuscript doesn’t deserve to be published in the present format, there are some aspects to be clarified.
General comments
-The authors deal extensively with the different varieties of Scorzanera, the secondary metabolites isolated and the biological activities associated with the extracts. However, the part dedicated to the experimental study remains rather modest. This is due to the fact that the evaluation of the volatile components has already been carried out by Yayh et al 2021 (DOI: http://doi.org/10.25135/rnp.240.21.02.1980 ) on S. pisidica and S. sandrasica.
- Because of the way it is set up, it looks more like a review article than a research article.
- An interesting review can already be found in the literature (not evaluated by the authors) https://doi.org/10.1016/j.jep.2023.116787
- Table 1 should indicate: the part of the plant containing the secondary metabolites, whether the biological tests were carried out on extracts or on purified fractions.
- Why the authors divided Table 1 into 3 parts.
- I think it would be appropriate to add the chemical structures of the compounds identified by GC-MS and to add the GC-MS TIC chromatograms of the 3 parts analysed (root, leaves and seeds).
Specific comments
In the key words add …volatile compounds
Line 74 Table 1 : (see above)
Line 86: Why the authors divided Table 2 into 3 parts. Specify
Table 2: Kovats indices (RI) are shown but experimental retention indices are not shown. Add these data.
Line 96 Figure 1 : Check the legend because it does not match.
Line 280-284: specify mass spectrometry (MS) detector and the ionization voltage.
Generally, the identification of components is based on the comparison of their Kovats retention indices (Exp RI), determined in relation to the tR values of a homologous series of n-alkanes (C8–C20). Add these data in Materials and Methods.
Line 292: In the conclusions, in the light of ethnobotanical knowledge, state what benefits can be derived from this study.
Comments on the Quality of English Language
The English could be improved
Author Response
The manuscript reports the volatile secondary metabolites occurring in different parts of the plant S. pseudolanata harvested in Turkey. Extraction was performed by SPME at room temperature followed by GC-MS analysis. The novelty and originality of this manuscript is fair, but there are interesting data to be a contribution to the chemotaxonomy of S. pseudolanata.
The manuscript doesn’t deserve to be published in the present format, there are some aspects to be clarified.
General comments
-The authors deal extensively with the different varieties of Scorzanera, the secondary metabolites isolated and the biological activities associated with the extracts. However, the part dedicated to the experimental study remains rather modest. This is due to the fact that the evaluation of the volatile components has already been carried out by Yayh et al 2021 (DOI: http://doi.org/10.25135/rnp.240.21.02.1980 ) on S. pisidica and S. sandrasica.
- Because of the way it is set up, it looks more like a review article than a research article.
To underline the importance of that this is the first report regarding voletile oil composition of S. pseudolanata additionally theused parts of every investigated Scorzonera species was added.
- An interesting review can already be found in the literature (not evaluated by the authors) https://doi.org/10.1016/j.jep.2023.116787
This review was also included to the manuscript.
- Table 1 should indicate: the part of the plant containing the secondary metabolites, whether the biological tests were carried out on extracts or on purified fractions.
The used plant parts was added.
- Why the authors divided Table 1 into 3 parts.
Corrected as one Table
- I think it would be appropriate to add the chemical structures of the compounds identified by GC-MS and to add the GC-MS TIC chromatograms of the 3 parts analysed (root, leaves and seeds).
The chemical structures of identified compunds are given as supplementary material.
Also the obtained chromatogramm is added to supplementary material.
Specific comments
In the key words add …volatile compounds
Was added
Line 74 Table 1 : (see above)
Was corrected.
Line 86: Why the authors divided Table 2 into 3 parts. Specify
Was corrected as one part
Table 2: Kovats indices (RI) are shown but experimental retention indices are not shown. Add these data.
Was added as supplementary data
Line 96 Figure 1 : Check the legend because it does not match.
The legedn weas corrected.
Line 280-284: specify mass spectrometry (MS) detector and the ionization voltage.
Are given below
Ionization voltage : 70 V
Generally, the identification of components is based on the comparison of their Kovats retention indices (Exp RI), determined in relation to the tR values of a homologous series of n-alkanes (C8–C20). Add these data in Materials and Methods.
Was added.
https://www.sigmaaldrich.com/TR/en/product/supelco/49451u?srsltid=AfmBOoo8-iC0yfsQDs5sLuo7ykz1pD7oDPOsWyIRzuuLWYekzRBjqPkd
Line 292: In the conclusions, in the light of ethnobotanical knowledge, state what benefits can be derived from this study.
Was corrected as “This study represents the first investigation into the secondary volatile metabolite composition of Scorzonera pseudolanata. The preliminary results revealed notable differences in the secondary metabolite profiles among the various plant parts of this species. However, further research should validate these findings using additional S. pseudolanata samples. The data presented here provide a valuable foundation for future, more comprehensive studies and contribute to a deeper understanding of the chemical composition of S. pseudolanata."
Reviewer 3 Report
Comments and Suggestions for Authors
The present article reports on the GC-MS analysis of the volatile constituents of the roots, leaves and seeds of the medicinal plant Scorzonera pseudolanata; this is the first study of the volatiles of this plant. The results are of interest; however, there are some points which need to be addressed by the Authors:
- Introduction, subsection ”Biological Activities in Scorzonera Species”: This part is inappropriate in a research article; this is not a review article. The Authors have to mention briefly these activities, especially those connected to essential oils.
- Line 72: Table 1 does not contain any biological data.
- Table 2, the caption: %of what? Of TIC? Or was it determined by GC with FID? Please mark it here.
- Lines 112 – 123: This information is in the Table 2, it should not be repeated in the text.
- Lines 143 – 182: this part has to be significantly shortened, such detailed description of results concerning other species is inappropriate.
English Language needs significant improvement.
Author Response
The present article reports on the GC-MS analysis of the volatile constituents of the roots, leaves and seeds of the medicinal plant Scorzonera pseudolanata; this is the first study of the volatiles of this plant. The results are of interest; however, there are some points which need to be addressed by the Authors:
- Introduction, subsection ”Biological Activities in Scorzonera Species”: This part is inappropriate in a research article; this is not a review article. The Authors have to mention briefly these activities, especially those connected to essential oils.
The title “Biological Activites etc.” was removed.
To give an better overwiew the used plant parts was added to Table 1. Of course, this is not an review article, but we have to underline that this is the first report about the volatile oil composition of S. pseudolanata.
- Line 72: Table 1 does not contain any biological data.
Was added
- Table 2, the caption: %of what? Of TIC? Or was it determined by GC with FID? Please mark it here.
Was corrected in Method part.
- Lines 112 – 123: This information is in the Table 2, it should not be repeated in the text.
Was removed.
- Lines 143 – 182: this part has to be significantly shortened, such detailed description of results concerning other species is inappropriate.
This brief description is necessary, because the volatile oil composition of other Scorzonera species like S. acuminata, S. papposa and S. mollis was determined in different plant parts. The authors tried to underline the diversity of chemical components in S: pseudolanata compared with different Scorzonera species.
Round 2
Reviewer 2 Report
Comments and Suggestions for Authors
Based on the reviewers' suggestions, the manuscript has been thoroughly revised, however the manuscript still requires many changes and clarifications. It is important that the authors submit a readable version, as it is difficult to follow the text given the numerous changes that have been made.
General comments
Authors often use the terms 'volatile metabolites' and 'essential oil components'. Please choose the most appropriate term, as these are compounds obtained through two different extraction processes. SPME is used for volatile metabolites and hydrodistillation for essential oils.
Specific comments
TITLE modify in: Composition of volatile secondary metabolites in different parts of Scorzonera pseudolanata Grossh. L. from Turkey.
Keywords add:……secondary volatile metabolite, SPME
Line 90: Volatile components or essential oil : please clarify
Line 90-100: essential oils are not listed in references 13 and 17, please check.
Line 125 Legend : Distribution of the composition of volatile secondary metabolites in the roots, leaves and seeds of S. pseudolanata
Line 127 : Replace Table 2 with the Table in the Supplementary Information (Rivalidation).
Line 198-199 : Delete the sentence that has been repeated too many times in the text.
Line 191-322: When discussing the different varieties of Scorzonera and their identified natural products, I suggest always to indicate the geographical origin (Turkey or other and Mediterranean areas). This will help to better assess the differences/similarities.
Line 249 : …phytoene….is reported in the text. Please clarify, this component is not present in the Table 2.
Line 331-332: please check the sentence.
Line 393… material and methods, report Sample preparation in a separate paragraph.
Line 431…: Authors must include the full experimental section on the use of the Kovat’s index and the standards used in GC-MS analysis.
Line 456 ….Preliminary results…: The results of the analysis are not preliminary, they are final. What does this sentence mean?
In the SUPPLEMENTARY information, I suggest to delete both FFNSC3 files because they are already listed in the references.
Authors must include 3 GC-MS chromatograms (TIC) for roots, leaves and seeds. Only the one for seeds has been included in the Supplementary Information. In addition, the chromatograms must be clearly visible, therefore the intensity of the signals could be more intense.
Comments on the Quality of English LanguageThe English could be improved
Author Response
İs added as word file.

Reviewer 3 Report
Comments and Suggestions for Authors
The Authors have made almost all the requested changes and amendments. There is only one point: Please add to the caption of Table 2 explanation: % of what? Of TIC? Or was it determined by GC with FID? Please mark it here.
Author Response
İs added as word file.

Round 3
Reviewer 2 Report
Comments and Suggestions for Authors
The manuscript requires minor revisions.
The authors report as an answer: We didn’t understand this part.
I now clarify:
1-Line 65 …. Volatile oils exhibit various biological activities, including insecticidal, antiviral, antioxidant, and antibacterial properties [13]…: Reference 13 only reports on antioxidant and antibacterial activity, not antiviral and insecticidal activity. Add other appropriate references.
2-Line 71 REFERENCE 17 : Reference 17 gives the isolation of inulin from S. hispanica and there is no discussion of cosmetics or perfumes. Please Include a reference that agrees with the sentence in the manuscript.
-The sentence : This study represents the first investigation into the secondary volatile metabolite composition of Scorzonera pseudolanata…. is repeated too many times in the manuscript at lines 24, 85, 136 and 315. Delete sentence at line 85 and 136.
Comments on the Quality of English LanguageThere are many errors in the text
Author Response
Answer to reviewers 3
1-Line 65 …. Volatile oils exhibit various biological activities, including insecticidal, antiviral, antioxidant, and antibacterial properties [13]…: Reference 13 only reports on antioxidant and antibacterial activity, not antiviral and insecticidal activity. Add other appropriate references.
The reference "Ben Miri Y. Essential Oils: Chemical Composition and Diverse Biological Activities : A Comprehensive Review. Natural Product Communications. 2025;20(1). " was added in olace of “Guclu, G.; Eruygur, N.; Ucar, E.; Ozbek, D.U.; Bal, O.; Akpulat, H.A.; Kahrizi, D. Biological Activity Evaluation of Scorzonera tomentosa L. Journal of Turkish Agricultural Research 2023, 10(2), 162–167.” Which is now reference 45
2-Line 71 REFERENCE 17 : Reference 17 gives the isolation of inulin from S. hispanica and there is no discussion of cosmetics or perfumes. Please Include a reference that agrees with the sentence in the manuscript.
This reference is adopted from Sharma; A.; Kumar;, V.; Mittal; C.; Rana; V.; Dabral; K.; Parveen; G.. Role of essential oil used pharmaceutical cosmetic product. Journal for Research in Applied Sciences and Biotechnology, 2023, 2(3), 147-157.
Therefore this is reference number 17 now.
The reference “Petkova, N. Characterization of inulin from black salsify (Scorzonera hispanica l.) for food and pharmaceutical purposes. Asian J. Pharm. Clin. Res. 2018, 11(12), 221-225.” İs now reference number 37
-The sentence : This study represents the first investigation into the secondary volatile metabolite composition of Scorzonera pseudolanata…. is repeated too many times in the manuscript at lines 24, 85, 136 and 315. Delete sentence at line 85 and 136.
sentence at line 85 and 136 were deleted.
All reference are ordered following their manuscript rank.
